# Sugar Beverage Habitation Relieves Chronic Stress-Induced Anxiety-like Behavior but Elicits Compulsive Eating Phenotype via vLS^GAD2^ Neurons

**DOI:** 10.3390/ijms24010661

**Published:** 2022-12-30

**Authors:** Dan Liu, Haohao Hu, Yuchuan Hong, Qian Xiao, Jie Tu

**Affiliations:** 1Shenzhen Key Laboratory of Neuroimmunomodulation for Neurological Diseases, Shenzhen-Hong Kong Institute of Brain Science, Shenzhen Institute of Advanced Technology, Chinese Academy of Sciences, Shenzhen 518055, China; 2CAS Key Laboratory of Brain Connectome and Manipulation, The Brain Cognition and Brain Disease Institute (BCBDI), Shenzhen Institute of Advanced Technology, Chinese Academy of Sciences, Shenzhen 518055, China; 3Guangdong Provincial Key Laboratory of Brain Connectome and Behavior, The Brain Cognition and Brain Disease Institute (BCBDI), Shenzhen Institute of Advanced Technology, Chinese Academy of Sciences, Shenzhen 518055, China; 4University of Chinese Academy of Sciences, Beijing 100049, China; 5Faculty of Life and Health Sciences, Shenzhen Institute of Advanced Technology, Chinese Academy of Sciences, Shenzhen 518055, China

**Keywords:** CUMS, anxiety, compulsive eating, vLS, GABAergic neurons

## Abstract

Chronically stressed individuals are reported to overconsume tasty, palatable foods like sucrose to blunt the psychological and physiological impacts of stress. Negative consequences of high-sugar intake on feeding behavior include increased metabolic disease burdens like obesity. However, the neural basis underlying long-term high-sugar intake-induced overeating during stress is not fully understood. To investigate this question, we used the two-bottle sucrose choice paradigm in mice exposed to chronic unpredictable mild stressors (CUMS) that mimic those of daily life stressors. After 21 days of CUMS paralleled by consecutive sucrose drinking, we explored anxiety-like behavior using the elevated plus maze and open field tests. The normal water-drinking stressed mice displayed more anxiety than the sucrose-drinking stressed mice. Although sucrose-drinking displayed anxiolytic effects, the sucrose-drinking mice exhibited binge eating (chow) and a compulsive eating phenotype. The sucrose-drinking mice also showed a significant body-weight gain compared to the water-drinking control mice during stress. We further found that c-Fos expression was significantly increased in the ventral part of the lateral septum (vLS) of the sucrose-treated stressed mice after compulsive eating. Pharmacogenetic activation of the vLS glutamate decarboxylase 2(GAD2) neurons maintained plain chow intake but induced a compulsive eating phenotype in the naïve GAD2-Cre mice when mice feeding was challenged by flash stimulus, mimicking the negative consequences of excessive sucrose drinking during chronic stress. Further, pharmacogenetic activation of the vLS^GAD2^ neurons aggravated anxiety of the stressed GAD2-Cre mice but did not alter the basal anxiety level of the naïve ones. These findings indicate the GABAergic neurons within the vLS may be a potential intervention target for anxiety comorbid eating disorders during stress.

## 1. Introduction

Chronic stress induces changes in circadian rhythm, anhedonia, anxiety, and behavioral despair in humans [1]. Stress-related neural circuits are activated during stress to orchestrate brain and body responses to minimize homeostatic disruption and promote survival. Stress also has significant impacts on feeding behavior [2]. Stress has clear effects on mood including increased anxiety, for many individuals, stress promotes the consumption of highly palatable and high-sugar foods in response to negative emotions such as anxiety [3]. Consuming high-sugar foods can attenuate the psychological (anxiety and depressive mood) and physiological effects of stress [4], but studies have also shown that excessive sugar beverage consumption can affect feeding behavior [5] and overeating, leading to increased prevalence of obesity and diabetes [6,7,8,9]. Despite clear functional associations among stress, food intake, energy balance, and emotion, the neuronal mechanisms underpinning these outcomes remain poorly understood.

The lateral septum (LS) plays a key role in emotional processes and stress responses [10]. The LS is located in the ventral medial of the lateral ventricle, sharing a border with the nucleus accumbens (NAc) and medial septal nucleus (MS). The LS directly projects to the NAc, bed nuclei of the stria terminalis (BNST), amygdala (AMG), hypothalamus, ventral tegmental area (VTA), thalamus, and pontine central gray [11,12]. Reciprocally, the LS receives inputs from the amygdala, hippocampus, hypothalamus, thalamus, VTA, midbrain, and hindbrain [12,13]. Neural circuits involving the LS are implicated in aggression [13,14,15], anxiety [16], and feeding [17], especially hedonic feeding [18]. In rodents, the LS contains dorsal (dLS), intermediate (iLS), and ventral parts (vLS) [19]. Anatomical studies suggest that the dLS contains GABAergic neurons that send inhibitory projections to the vLS [20,21]; these GABAergic neurons in the dLS and iLS maintain wakefulness and accelerate recovery from anesthesia [22]. Recently, Bales et al. [23] revealed that dLS^Glp1r^ neuronal activation promotes hypophagia in response to acute restraint stress in male mice, but not in female mice. It has also been reported that the vLS sends inhibitory GABAergic projections to the ventrolateral subnucleus of the ventromedial hypothalamus (vlVMH) to regulate aggression [24], with this subnucleus also associated with anxiety-like behavior [25]. Intriguingly, the dominant neuronal population in the vLS is GABAergic [26]. In humans, variations in glutamate decarboxylase 2 (GAD2) are associated with feeding behaviors, emotional eating, and weight gain [27]. However, the specific functions of vLS GABAergic neurons in linking the outcomes of emotional regulation and feeding during stress are not fully understood.

In this study, we used the two-bottle sucrose choice paradigm in mice under chronic unpredictable mild stress (CUMS) to mimic sugar beverage habits under stress. We found that the sucrose-consuming group showed lower anxiety-like behavior during CUMS than the water-consuming group. However, the sucrose-consuming group also showed a compulsive eating phenotype accompanied by activation of GABAergic neurons in the vLS. As a negative consequence, body-weight gain was significantly increased in the sucrose-drinking group compared to the water-drinking controls. Pharmacogenetic activation of vLS^GAD2^ neurons in GAD2-cre mice elicited compulsive eating regardless of negative flash stimulus, thus mimicking the negative consequences of excessive sucrose intake during chronic stress. Meanwhile, pharmacogenetic activation of the vLS^GAD2^ neurons aggravated anxiety of the stressed GAD2-Cre mice but did not alter the basal anxiety level of the naïve ones. Our findings highlight a potential neuronal target for therapeutic intervention in anxiety comorbid eating disorders.

## 2. Results

### 2.1. CUMS-Induced Anxiety-like Behavior

Anxiety-like behaviors were tested using open field test (OFT) and elevated plus maze (EPM) after 21 days of exposure to random daily stressors (Figure 1A). For OFT, less time spent in the central area represents a higher level of anxiety (Figure 1B–E). Here, results showed that the CUMS mice spent less time (t = 3.218, *p* = 0.0050) and exhibited fewer entries (t = 3.005, *p* = 0.0080) in the center area compared to control mice, without a change in mean speed (t = 1.664, *p* = 0.1145). For the EPM test (Figure 1F–I), less time spent in the open arms represents a higher level of anxiety. Here, results showed that the CUMS mice spent less time (t = 3.008, *p* = 0.0083) and exhibited fewer entries (t = 2.549, *p* = 0.0214) in the open arms compared to control mice, without a change in mean speed (t = 1.419, *p* = 0.1752). These results suggest that CUMS induces anxiety-like behavior in C57BL/6J mice.

### 2.2. Sucrose Solution Caused an Anxiolytic Effect

We next investigated whether sucrose could relieve anxiety-like behavior in stressed mice. After finding that CUMS induced anxiety-like behaviors, we paired increasing sucrose concentrations (up to 15%) with CUMS in mice (Figure 2A) and performed the same behavioral tests. Results showed no difference between the CUMS + water and CUMS + sucrose groups in mean speed (Figure 2B, t = 1.117, *p* = 0.2785) in the OFT. Compared with the CUMS + water group, the CUMS + sucrose group spent more time (Figure 2C, t = 2.514, *p* = 0.0217) in the center area and showed less freezing over 300 s (Figure 2D, t = 2.856, *p* = 0.0105). The heat map of the OFT representative mice was shown as Figure 2E, red color means mice spent more time in that place. For the EPM test (Figure 2F–I), the CUMS + sucrose mice spent more time (t = 3.196, *p* = 0.0050) and exhibited more entries (t = 2.858, *p* = 0.0105) in the open arms compared to CUMS mice, without a change in mean speed (t = 0.9442, *p* = 0.3576). The heat map of the OFT representative mice was shown as Figure 2E, red color means mice spent more time in that place. These results suggest that CUMS-induced anxiety-like behavior can be reduced by sucrose intake.

### 2.3. Sucrose Concentrations Induced Body-Weight Gain by Increasing Total Caloric Intake

C57BL/6J mice were exposed to 21 consecutive double-bottle cage, CUMS + water group choose water/water freely, and sucrose group choose water/sucrose solution freely (Figure 3A). Total caloric intake (Figure 3B) was analyzed by two-way repeated ANOVA, and multiple comparisons showed that total caloric intake was increased by 29% (t = 3.441, *p* = 0.0284) on day 16. Greater caloric intake induced body weight gain (Figure 3C) on day 16 (9%, t = 4.202, *p* = 0.0004) and day 20 (8%, t = 3.685, *p* = 0.0025). Surprisingly, mice treated with sucrose solution consumed less chow (Figure 3D) than the CUMS + water group on day 8 (t = 3.679, *p* = 0.0298). Total caloric intake from sucrose solution increased on day 12 compared with day 8 (Figure 3E, q = 7.460, *p* < 0.0001). Thus, these results suggest that sucrose solution suppresses food intake but increases total caloric intake and weight gain.

### 2.4. Sucrose Solution-Induced CPP

Before the DBT, a pre-test was carried out to determine the baseline of time traveled around a neutral cue without sucrose solution (Figure 4A). After 20 days of training, mice learned that this cue was paired with a sucrose solution. A post-test was conducted without sucrose solution on day 20 to measure sucrose solution-induced CPP. The four quadrants were marked as with sucrose solution (S), with water (W), blank quadrant no sucrose (NS), and blank quadrant no water (NW). Time spent in the four quadrants was recorded to evaluate preference. Results showed no difference in time spent in the four quadrants between the CUMS + water group and CUMS + sucrose group (Figure 4B). After training, however, the CUMS + sucrose group spent more time in the cue quadrant (Figure 4C, t = 2.777, *p* = 0.01483). These findings suggest that sucrose solution can induce CPP-like behavior as a natural reward.

### 2.5. Binge Phenotype in Sucrose-Treated Group

After 21 days of sucrose exposure, feeding patterns in the water- and sucrose-treated groups were measured by chow intake (Figure 5A). After 12 h of fasting, chow intake was measured for 1 h to test binge eating. Results indicated that compared with the water-treated group, the sucrose-treated mice consumed more standard chow within 1 h (Figure 5B, t = 3.016, *p* = 0.0082), thus displaying binge eating behavior. This may be another characteristic of food addiction called cross-sensitization. Under natural circumstances, flashing light can act as a negative stimulus to suppress feeding. Here, compulsive eating was defined as eating despite negative stimuli. We measured chow intake (Figure 5C, Feeding Factor MS = 2.951, *p* = 0.0007) under flashing light (Flash Factor MS = 5.325, *p* < 0.0001), which partially stopped feeding behavior in the CUMS group (t = 4.123, *p* = 0.0005), but did not stop feeding behavior in the binge (sucrose-treated) group (t = 1.006, *p* = 0.5405). Thus, these results suggest that sucrose intake can induce binge and compulsive eating phenotypes.

### 2.6. c-Fos Showed Different Activation in the vLS after Binging

Based on whole-brain c-Fos mapping data (Figure 6A), c-Fos-positive cells in the anterior olfactory nucleus medial part (AOM, t = 3.784, *p* = 0.000598), accumbens nucleus shell (NAcS, t = 2.352, *p* = 0.02462), and vLS (t = 3.669, *p* = 0.000828) were differentially increased in the sucrose-treated stressed mice after binging on chow. Of note, c-Fos expression in the vLS showed a 55% increase compared to that in the water-treated stressed mice (Figure 6B–D). The c-Fos mapping results in the AOM, dLS, and anterior basolateral amygdaloid nucleus (BLA) are shown in Figure 6E–G. After the binge and compulsive eating tests, c-Fos expression in the vLS was up-regulated, suggesting that the vLS may be involved in compulsive eating behavior.

### 2.7. Pharmacogenetic Activation of vLS^GAD2^ Neurons Induced Compulsive Eating Phenotype

The Gad2 gene is widely expressed in the vLS (Allen Brain) and variations in GAD2 are associated with eating behaviors, including emotional eating, and weight gain in women [27]. GAD2-Cre mice were used to perform pharmacogenetic activation experiments. After stable expression of AAV-DIO-hM3D(Gq)-mCherry or AAV-DIO-mCherry in the vLS, GAD2-Cre mice were injected (i.p.) with 2 mg/kg CNO to activate the hM3D(Gq) component expressed in vLS^GAD2^ neurons prior to behavioral testing (Figure 7A). CNO administration did not induce the binge eating phenotype in the naïve mice (Figure 7B, t = 0.5477, *p* = 0.5988). However, when the GAD2-Cre mice were challenged by negative flashlight, pharmacogenetic activation vLS ^GAD2^ neurons induced compulsive eating in the hM3D(Gq) group (Figure 7C, t = 0.7950, *p* = 0.6844), manifested as food cessation in the mCherry group (Figure 7C, t = 3.081, *p* = 0.0143). Interestingly, pharmacogenetic activation of vLS ^GAD2^ neurons did not alter basal anxiety level compared with the mCherry group in the EPM test (Appendix A), whereas pharmacogenetic activation of vLS ^GAD2^ neurons aggravated anxiety-like behavior in OFT after exposure to the social defeat stress (Figure 7D–I). In OFT, time spent (Figure 7E, t = 2.856, *p* = 0.0189), distance traveled in the center area (Figure 7F, t = 2.754, *p* = 0.0223) as well as entries to the center area (Figure 7G, t = 3.092, *p* = 0.0129) of hM3Dq group was less than mCherry group, whereas the freezing time (Figure 7H, t = 2.828, *p* = 0.0198) and immobile time (Figure 7I, t = 2.706, *p* = 0.0241) was increased compared to the mCherry group, which indicated that pharmacogenetic activation of vLS ^GAD2^ aggravated anxiety-like behavior induced by stress. Overall, activation of vLS^GAD2^ neurons induced a compulsive eating phenotype, and induced an anxiety-like behavior detected by OFT after stress.

## 3. Discussion

In our study, we verified that CUMS induced anxiety-like behavior in mice, which was alleviated by drinking sucrose. However, total caloric intake, body weight, and sucrose consumption were higher in the sucrose-treated group than in the water-treated control. Mice successfully demonstrated CPP for sucrose solution after training. Importantly, after 21 days of CUMS concurrent with sucrose consumption, mice displayed binge eating behavior and exhibited a compulsive eating phenotype. Furthermore, c-Fos expression was significantly increased in the vLS of the sucrose-treated stressed mice after binge eating chow. Pharmacogenetic activation of vLS ^GAD2^ neurons induced a compulsive eating phenotype, thus mimicking the negative consequences of hedonic eating during stress.

### 3.1. Opposite Effects of CUMS and Sucrose on Negative Emotions

CUMS can simulate a variety of stresses in human daily life. In the current study, mice were subjected to daily random stress, including no sawdust, damp sawdust, changes in sawdust, social stress, restraint, fasting, and water deprivation. This unpredictable negative stress kept mice in a constant state of anxiety. In our previous study [28] and Yue et al. [29], CUMS induced both depression- and anxiety-like behaviors, manifested as abnormalities in OFT and EPM. Although CUMS induced anxiety-like behavior in the present research, it did not induce depression-like behavior. Thus, we believe that 3 weeks of mild stimulation is sufficient to induce anxiety but not depression, while 4 weeks of CUMS is sufficient to induce both [30]. In this study, sugar water was given at the same time as CUMS stimulation, so the classic sucrose preference test (SPT) was not applicable for detecting depression-like behaviors.

Psychological and social distress, as experienced during the COVID-19 pandemic and lockdown, can lead to emotional eating [31], a potential protective mechanism for calming negative emotions. High-sucrose intake has been shown to relieve anxiety and the physiological (hypothalamus-pituitary-adrenal gland axis) effects of stress [4,32]. Previous research has also found that immediate sucrose rewards can alleviate anxiety-like behavior during acute stress and inhibit paraventricular corticotropin-releasing hormone (CRH) neurons [33]. Recent research has also found that sucrose consumption can alleviate depression-like behavior in flies by increasing serotonin levels [34]. Notably, in our study, consumption of sucrose relieved anxiety-like behavior in chronically stressed mice. These findings suggest that, as a reward, sucrose may alleviate anxiety and depression caused by acute and chronic stress.

In our study, 21 days of CUMS paired with sucrose solution intake produced an anxiolytic effect in mice. Unexpectedly, however, the sucrose-treated group also showed binge eating behavior and a compulsive eating phenotype. Excessive consumption of sugar has been reported to alter feeding behavior, including overeating [5]. Many metabolic diseases, such as diabetes, hypertension, and obesity, as well as neurological disorders, are increased as a negative consequence of overeating [6,9,35]. However, further research is needed to understand the mechanisms underlying the associations among stress, food intake, energy balance, and emotional regulation.

### 3.2. Inverted U-Shape of Stress and Appetite Relationship

In humans, mild stress can lead to increased appetite, while disaster stress can lead to decreased appetite [36]. We hypothesize that the relationship between stress and appetite appears as an inverted U-shape, with the apex of the U-shape dependent on our tolerance to negative stimuli. Negative stress can be classified as mild, moderate, or severe. We speculate that emotional eating during periods of mild stress may be a form of self-relief but may also lead to an increase in appetite. In the current study, we used chronic mild stress to induce anxiety-like behavior in mice and found that sucrose consumption produced anxiolytic effects in the stressed mice. However, humans and animals respond differently to moderate and severe stimuli. For example, feeding behavior is reduced in rats subjected to inescapable shock [37] and appetite is suppressed in humans under acute stress [38]. Therefore, further studies are required to identify the distinct neural circuits that mediate different feeding motivations in response to different stress intensities.

We proposed that the relationship between stress and appetite manifests as an inverted U-shape. In this study, we found that sucrose consumption relieved anxiety-like behavior in mice but also induced binge eating of normal chow. We interpret these results as a horizontal shift in the vertices of the inverted U-shaped curve. This transition can be described as an inverted U-shaped apex shift, with anorexia defined by a shift in the apex of the inverted U-shaped curve to the left, and food addiction and overeating defined by a shift to the right.

### 3.3. Energy Imbalance in Liquid and Solid Sucrose

Balancing total caloric intake is difficult when drinking sugar-sweetened beverages (SSBs). In an epidemiological survey of adults in the United States, total caloric intake was higher among normal weight humans who consume SSBs compared with a control diet [39]. Similarly, our findings showed that total caloric intake was higher in the sucrose-consuming group than in the water-consuming group. These additional calories came from the sucrose solution rather than solid normal chow and resulted in a significant increase in body weight in the mice. Although the sucrose-treated mice consumed less solid chow to balance total intake during the first stage of the experiment, they failed to control their body weight and total caloric intake at the later stage of the experiment, suggesting that feeding balance was disrupted by sucrose (15%) consumption. A 15% sucrose dose is equivalent to a 2.2% sucrose beverage, which is exceeded by most beverages in our daily lives. Consistent with our findings, previous research has also reported that consumption of sucrose solution, but not equivalent levels of solid sucrose, results in body fat gain in C57BL/6 mice [40]. However, whether and how sugar consumption blunts the brain reward system, resulting in overeating and energy imbalance, requires further investigation.

### 3.4. Involvement of vLS^GAD2^ Neurons in Emotional/Compulsive Eating and Anxiety

Emotional eating is defined as eating for pleasure. A recent study identified a potential neural circuit specifically involved in hedonic feeding regulation, in which neurotensin (Nts) neurons in the LS project to the tuberal nucleus (TU) via GABA signaling to regulate hedonic feeding, with chronic activation of this circuit sufficient to reduce high-fat diet-induced obesity [18]. Coincidentally, a previous human population study indicated that GAD2 variations are associated with feeding behavior, weight gain, and emotional eating [27]. Therefore, we speculate that different types of neurons in the vLS are involved in feeding behaviors, ranging from homeostatic feeding, emotional feeding, hedonic feeding, compulsive feeding, to food addiction. In the addiction phase, subjects tend to show compulsive eating, regardless of the negative consequences [41], a recent study showed iLS-NAc circuit mediates stress induced suppression of natural reward seeking [42]. Our results indicated that compulsive eating activated the vLS, implying that the vLS may be involved in regulating the plasticity of the brain reward system. Our data further showed in the stressed mice, manifesting as the impaired reward and stress-response systems, activation of vLS GABAergic neurons enhanced anxiety levels, indicating these neurons may be the potential intervention target for the emotion disorders comorbid addiction. However, further studies are needed to verify this hypothesis.

## 4. Materials and Methods

### 4.1. Animals

GAD2-IRES-Cre (Jackson Laboratory, Stock No. 010802) and C57BL/6 male mice (6–8 weeks,18–22 g, Zhejiang Vital River Co., Jiaxing, China) were housed four per cage in pathogen-free conditions under a 12/12 h light/dark cycle, temperature of 22 ± 2 °C, and relative humidity of 50–60%, with free access to food and water. Animals were habituated to the facility environment for 1 week before experimentation. All studies were in compliance with the National Institutes of Health guide for the care and use of Laboratory animals (NIH Publications No. 8023, revised 1978) and approved by the Institutional Animal Care and Use Committee (IACUC, SIAT-IACUC-210201-NS-LD-A1539) at the Shenzhen Institute of Advanced Technology (SIAT), Chinese Academy of Sciences (CAS). Surgeries were performed under full anesthesia and every effort was made to minimize animal suffering.

### 4.2. Chronic Unpredictable Mild Stress (CUMS)

The CUMS protocol was performed as described previously [43,44] with some modification. Mice were daily exposed to random environmental stressors for 21 days as follows: A. Social stress (each mouse was placed in an empty cage previously occupied by another individual). B. Without sawdust for 12 h (bedding sawdust was removed from the home cage). C. Damp sawdust for 12 h (bedding sawdust was dampened in the home cage). D. Restraint (each mouse was placed in a tube (50 mL) for 2 h without access to food or water). E. Fasting (each mouse was deprived of food for 12 h). F. Water deprivation (each mouse was deprived of water for 12 h). G. Crowding (four mice were housed in a box (3 × 5 × 7 cm) for 2 h without access to food or water). H. Sawdust changes. For stress challenge in GAD2-Cre mice, aggressive CD1 mice co-house with clear acrylic sheet to separate the C57BL/6 and CD1 mice for 24 h, and behavior test was operated 30min later.

### 4.3. Double-Bottle Test (DBT)

To evaluate the preference for and consumption (quantity) of sucrose solution, C57 mice were divided into two groups: i.e., water-water group and water-sucrose group. Each test day, mice in the water-water group were housed separately in a home cage-like box containing two bottles of water [45,46]. For the water-sucrose group, mice were housed separately in a home cage-like box containing a bottle of water and a bottle of sucrose solution. The mice were given unlimited access to two bottles of water for 2 days before testing. From days 1 to 21, sucrose bottles were filled with increasing concentrations of sucrose (up to 15%) [47]. The position of the two bottles was changed every two days during the paradigm. Weight of liquid drinking was recorded, and calories were calculated.

### 4.4. Conditioned Place Preference (CPP)

To assess preference for sucrose solution in mice, we used a modified CPP (cue) test conducted in a home cage-like box (DBT box) to minimize stress. A. Pre-test: prior to the CUMS and DBT, mice were placed into a DBT box without a bottle. The box was paired with a rough card cue in one of the four quadrants, named the cue quadrant. Time spent in the cue quadrant was recorded and analyzed. B. Training: each day during DBT, the water-sucrose group was provided with a bottle containing sucrose solution paired with the cue, while the water-water group was provided with a bottle containing water paired with the cue. The cue-bottle position was changed every two days to exclude other factors. C. Post-test: after training, time spent around the cue was measured to evaluate conditioned place preference. The DBT box without a bottle but with the cue was used to test trained mice and the time spent (total 300 s) in the four quadrants was recorded. 

### 4.5. Elevated plus Maze (EPM) Test

The plastic EPM consisted of a central platform (5 × 5 cm) with two white open arms (25 × 5 × 25 cm) and two white closed arms (25 × 5 × 25 cm) extending from the center in a plus shape. The maze was elevated 65 cm above the floor. Mice were individually placed in the center with their heads facing a closed arm. The number of entries and amount of time spent in each arm were recorded for 300 s.

### 4.6. Open Field Test (OFT)

A plastic open field chamber (50 × 50 cm) was used and conceptually divided into a central field (25 × 25 cm) and a peripheral field for analysis. Each mouse was placed in the peripheral field at the start of each test. The number of entries and amount of time spent in the center were recorded for 300 s. 

### 4.7. Binge and Compulsive Eating Tests

After measuring their daily eating baseline, mice were separately fed to measure food intake for 1 h to evaluate food intake baseline. Binging was defined as eating more than 20% daily needs within 1 h [48]. After that, compulsive eating under flashing LED stress was detected for 1 h [49]. Compulsive eating was defined as eating despite negative consequences (such as flashing light stress). Chow intake in each experiment was recorded for 60 min and analyzed.

### 4.8. Brain Sample and Coronal Section Collection

Mice were deeply anaesthetized using isoflurane, and transcardially perfused with 5 mL of phosphate-buffered saline (PBS), followed by 25 mL of 4% paraformaldehyde (PFA) in PBS. Brains were extracted and fixed in 4% PFA overnight, then stored in 30% sucrose at 4 °C until cutting. Whole brain slices were collected at 30–40 μm thickness with a microtome (Leica SM 2010R, Leica, Weztlar, Germany) in PBS.

### 4.9. Immunofluorescent Staining

Brain sections were initially incubated for 60 min in 3% bovine serum albumin (BSA) in 0.3% triton-X100 PBS solution. The brain slices were incubated with rabbit anti-c-Fos primary antibodies (1:300, CST 2250S, Cell Signaling Technology, Danvers, USA) in 0.1% triton-X100 PBS solution for 24 h at 4 °C, then rinsed with PBS three times on a shaker. The slices were then incubated in Alexa Fluor (AF) 488-conjugated goat-anti-rabbit secondary antibodies (1:200, Jackson Immuno Research, West Grove, USA) for 90 min at room temperature. Finally, all slices were rinsed with PBS three times (10 min each time) on a shaker and maintained at 4 °C. The sections were counter-stained with 10 μM DAPI (D-9542, Sigma, St. Louis, MO, USA), then mounted on microscope slides and cover-slipped with anti-fluorescence quencher (0100-01, Southern Biotech, Birmingham, AL, USA). 

### 4.10. Whole-Brain c-Fos Mapping

Immunofluorescent slices were scanned using a fluorescent microscope under 10× objective (VS200, Olympus, Tokyo, Japan). Labeled neurons in selected sections were imaged with a confocal microscope (LSM 900, Carl Zeiss, Jena, Germany). Most images were obtained and analyzed using OlyVia and ZEN blue edition software v3.1. To outline specific brain regions, Photoshop CS6 (Adobe Systems Incorporated, San Jose, CA, USA) was used to count c-Fos-positive neurons.

### 4.11. Adeno-Associated Virus (AAV) Injection and Pharmacogenetic Activation of vLS^GAD2^ Neurons

Mice were anesthetized with isoflurane (1–4%) and placed on a stereotaxic frame (RWD68001, RWD, China). Surgeries were performed under aseptic conditions. No more than 200 nL of cre-dependent AAV vector AAV-DIO-hM3D(Gq)-mCherry was injected into the vLS (AP +1.1, ML −0.6, DV −3.8) of GAD2-Cre mice to activate vLS^GAD2^ neurons. The syringe (Hamilton 1700) was mounted on a motorized microinjector (Legato^®^ 130; KD Scientific, USA) operating on the stereotaxic frame at an injection rate of 50 nL/min. After each injection, the needle remained in situ for 15 min to minimize backflow along the needle. After 3 weeks, the mice were handled each day for three consecutive days in preparation for the behavioral tests. For the behavioral tests, the mice received a single intraperitoneal (i.p.) injection of 2 mg/kg of clozapine N-oxide (CNO) in saline or saline alone 30 min before testing.

### 4.12. Statistical Analysis

All statistical parameters for specific analyses are described in the appropriate figure legends. Mouse locations were monitored/tracked using ANY-maze software v7.1. All data are presented as mean ± standard error of the mean (SEM). Statistical significance was assessed using two-tailed Student’s *t*-tests or two-way analysis of variance (ANOVA) with GraphPad v9.0.( GraphPad Software, San Diego, CA, USA) Post hoc multiple comparison method for ANOVA was Tukey or Bonferroni. Differences were considered statistically significant at *p* < 0.05.

## 5. Conclusions

Studies have suggested that the LS is both anxiolytic and anxiogenic [10]. Interestingly, anxiety is also associated with excessive risk-avoidance behavior [50]. A recent study found higher c-Fos expression and lower mGlu2/3Rs expression in the vLS of anxiety-susceptible mice after social defeat stress [25]. Our results showed that pharmacogenetic activation of vLS^GAD2^ neurons elicited compulsive eating regardless of negative flash stimuli, and deteriorated anxiety in the stressed mice. Taken together, these findings indicate that vLS^GAD2^ neurons may be involved in both reward seeking and risk avoidance simultaneously. In other words, these neurons may be the potential intervention targets for addiction comorbid affective disorders. However, further studies are needed to identify the upstream and downstream circuits of the vLS^GAD2^ neurons that account for the associations between the emotion regulation and reward seeking during chronic stress.

## Figures and Tables

**Figure 1 ijms-24-00661-f001:**
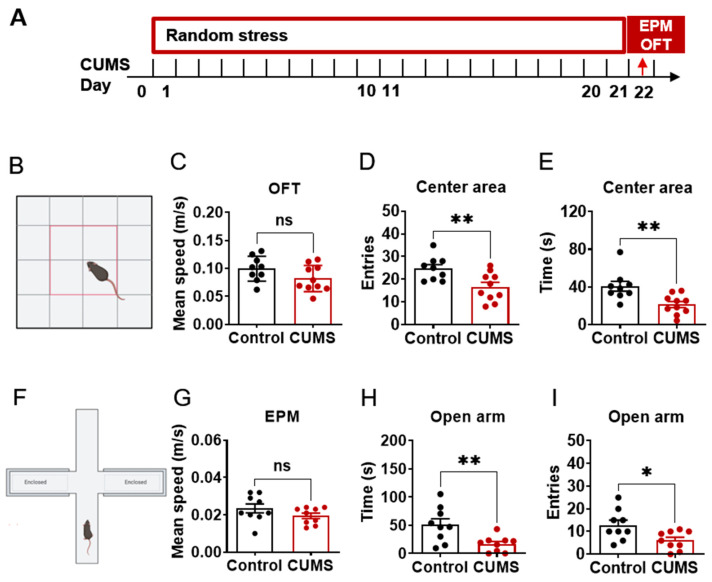
Chronic unpredictable mild stress (CUMS) induces anxiety like behavior. (**A**). Schematic of stress experiment, whereby mice were subjected to 21 days of random stress, including no sawdust, sawdust changes, social stress, damp sawdust, restraint, fasting, crowding, and water deprivation. (**B**). Schematic of open field test (OFT). Created with BioRender.com (Agreement number YB24KOMWEF). (**C**). Mean speed of mice in OFT. (**D**). Time spent in center square in OFT. (**E**). Number of entries into center square in OFT. (**F**). Schematic of elevated plus maze (EPM). Created with BioRender.com (Agreement number YB24KOMWEF). (**G**). Mean speed of mice in EPM. (**H**). Time spent in open arms in EPM. (**I**). Number of entries in open arms in EPM. N = 9–10. Data were analyzed by student *t*-test and are shown as mean ± SEM. ns, no significant difference. *, *p* < 0.05. **, *p* < 0.01.

**Figure 2 ijms-24-00661-f002:**
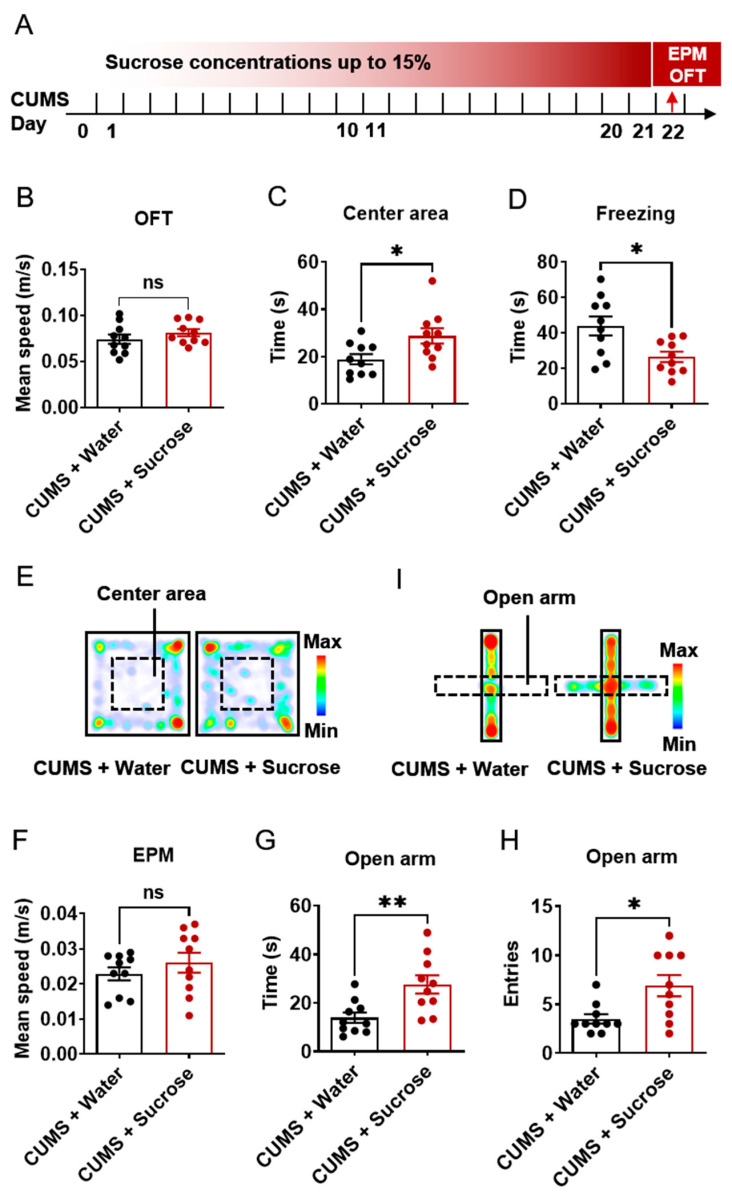
Sucrose solution consumption had an anxiolytic effect in stressed mice. (**A**). Schematic of increasing sucrose two-bottle solution treated under CUMS paradigm. (**B**). Mean speed of mice in OFT paradigm. (**C**). Time spent in center square in OFT. (**D**). Number of entries in center square in OFT. (**E**). Heat map of representative mouse trajectory in OFT. (**F**). Mean speed of mice in EPM paradigm. (**G**). Time spent in open arms in EPM. (**H**). Number of entries in open arms in EPM. (**I**). Heat map of representative mouse trajectory in EPM. N = 9–10. Data were analyzed by student *t*-test and are shown as mean ± SEM. ns, no significant difference. *, *p* < 0.05. **, *p* < 0.01.

**Figure 3 ijms-24-00661-f003:**
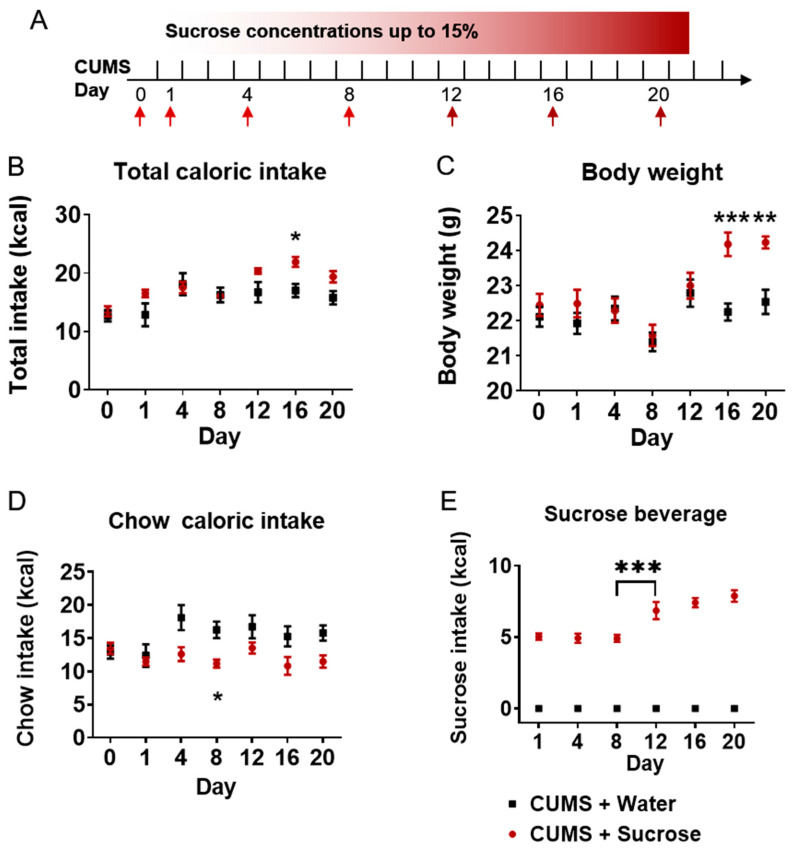
Sucrose induced body-weight gain by increasing total caloric intake. (**A**). Schematic of increasing sucrose two-bottle solution treated under CUMS paradigm. (**B**). Total caloric intake in sucrose solution-treated group and water-treated group under CUMS. (**C**). Body weight of sucrose solution-treated group and water-treated group under CUMS. (**D**). Chow caloric intake. (**E**). Sucrose solution caloric intake. N = 8–10. Data were analyzed by one-way repeated measures ANOVA and are shown as mean ± SEM. *, *p* < 0.05. **, *p* < 0.01. ***, *p* < 0.001.

**Figure 4 ijms-24-00661-f004:**
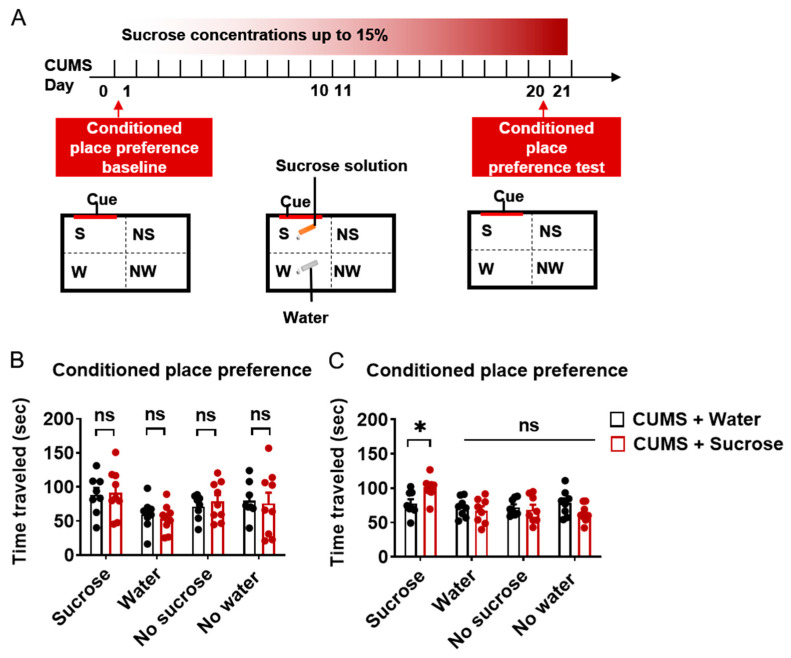
Sucrose solution induced conditioned place preference. (**A**). Schematic of cue-paired sucrose solution-induced cue preference. (**B**). Time traveled baseline of mice in a home cage-like box in four quadrants. (**C**). Time traveled after sucrose solution training of mice in a home cage-like box in four quadrants. N = 8–10. Data were analyzed by student *t*-test and are shown as mean ± SEM. ns, no significant difference.*, *p* < 0.05.

**Figure 5 ijms-24-00661-f005:**
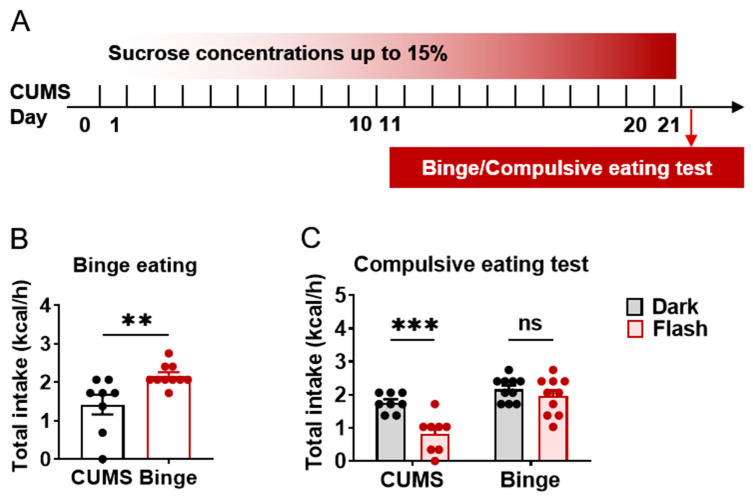
Binge phenotype observed in sucrose-treated group. (**A**). Schematic of cue-paired sucrose solution-induced cue preference. (**B**). Total chow intake of C57BL/6J mice within 1 h. (**C**). Total chow intake of C57BL/6J mice under dark or flashing light to measure compulsive eating. N = 8–10. Data were analyzed by student *t*-test and two-way ANOVA and are shown as mean ± SEM. ns, no significant difference, **, *p* < 0.01. ***, *p* < 0.001.

**Figure 6 ijms-24-00661-f006:**
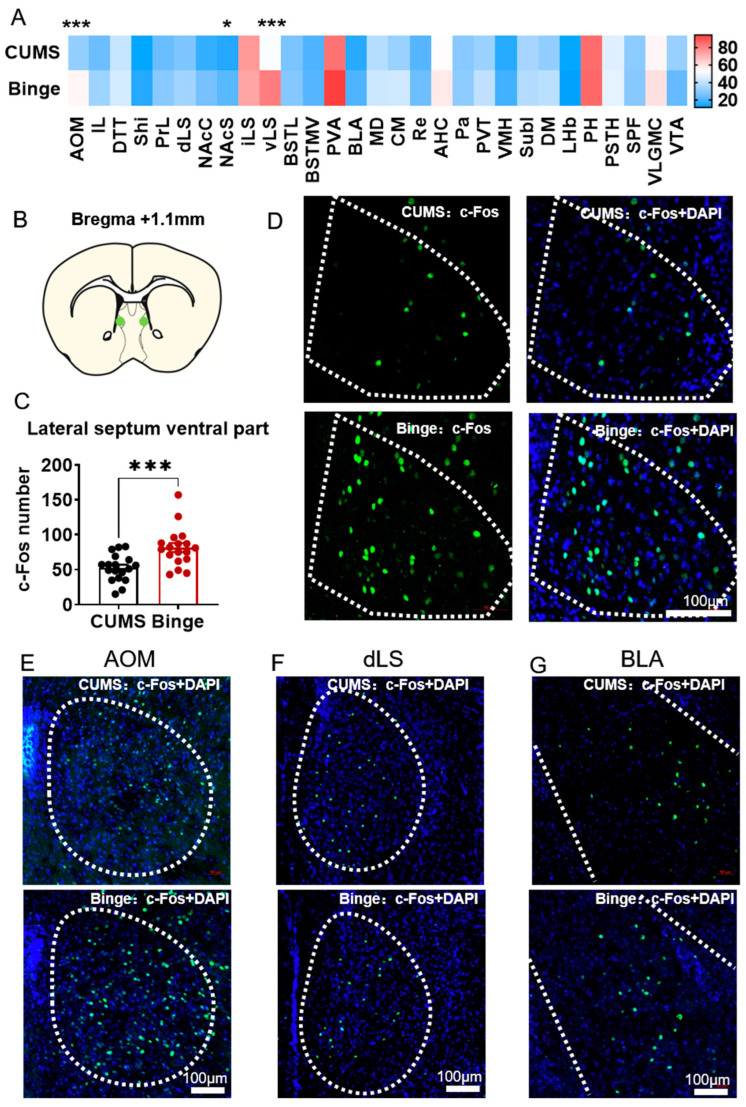
c-Fos showed different activation in lateral septum ventral part (vLS) after binging. (**A**). c-Fos mapping of brain regions in binging mice. (**B**–**D**). Sketch map of c-Fos-positive cells in vLS. (**E**–**G**). c-Fos-positive cells in AOM, dLS, and BLA. N = 3. Data were analyzed by student *t*-test and two-way ANOVA and are shown as mean ± SEM. *, *p* < 0.05. ***, *p* < 0.001. Abbreviations: AOM anterior olfactory nucleus medial part, IL infralimbic cortex, DTT dorsal tenia tecta, SHi septohippocampal nucleus, PrL prelimbic cortex, dLS lateral septal nucleus dorsal part, NAcC accumbens nucleus core, NAcSh accumbens nucleus shell, iLS lateral septal nucleus intermediate part, vLS lateral septal nucleus ventral part, BSTL bed nucleus of the stria terminalis lateral division, BSTMV bed nucleus of the stria terminalis medial division ventral part, PVA paraventricular thalamic nucleus anterior part, BLA basolateral amygdaloid nucleus anterior part, MD mediodorsal thalamic nucleus, CM central medial thalamic nucleus, Re reuniens thalamic nucleus, AHC anterior hypothalamic area central part, Pa paraventricular hypothalamic nucleus, PVT paraventricular thalamic nucleus, VMH ventromedial hypothalamic nucleus, SubI subincertal nucleus, DM dorsomedial hypothalamic nucleus, LHb lateral habenular nucleus, PH posterior hypothalamic area, PSTH parasubthalamic nucleus, SPF subparafascicular thalamic nucleus, VLGMC ventral lateral geniculate nucleus magnocellular part, VTA ventral tegmental area.

**Figure 7 ijms-24-00661-f007:**
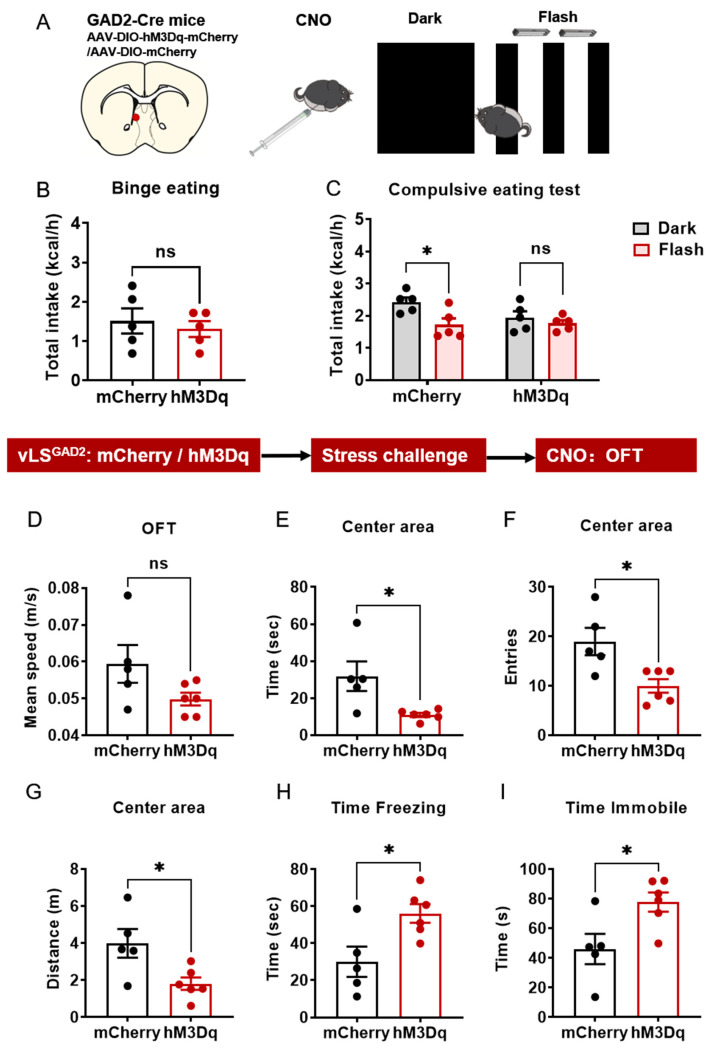
Pharmacogenetic activation of vLS ^GAD2^ neurons induced compulsive eating phenotype and enhanced anxiety level in the stressed mice. (**A**)**.** Schematic of chemical activation of vLS ^GAD2^ neurons and behavioral test. (**B**). Total chow intake of GAD2-cre mice within 1 h. (**C**). Total chow intake of GAD2-cre mice under dark or flashing light to measure compulsive eating. (**D**–**I**). Mean speed, time spent, number of entries and distance traveled in center square, freezing time and immobile time in OFT after stress and pharmacogenetic activation vLS ^GAD2^ neurons. N = 5–6. Data were analyzed by student *t*-test and two-way ANOVA and are shown as mean ± SEM. ns, no significant difference. *, *p* < 0.05.

## Data Availability

Not applicable.

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
