# Peer review of "Sugar Beverage Habitation Relieves Chronic Stress-Induced Anxiety-like Behavior but Elicits Compulsive Eating Phenotype via vLSGAD2 Neurons"

_ijms, 2022, doi:10.3390/ijms24010661_

Round 1
Reviewer 1 Report
In this study, Liu et al. investigated the effects of sugar solution consumption on chronic stress-induced anxiety-related behaviors, and explored the neuronal mechanisms behind such influence. They found that chronic unpredictable stress increased anxiety level in mice, which could be attenuated by sucrose solution consumption. However, sucrose-drinking mice also showed binge and compulsive eating. Using c-fos mapping and chemogenetics, they showed that sucrose consumption in stressed mice increased neuronal activation in the ventral part of the lateral septum, whereas chemogenetic activation of inhibitory interneurons in this region further increased anxiety level in stressed mice. These findings are novel and important to the field. The manuscript is generally well written and figures are well prepared. Nonetheless, I have the following comments that require attention.
Major issues:
· Strictly speaking, control groups in many experiments are missing, e.g., the two unstressed groups in figures 2-7. The rationale for using such a simplified experimental design should be clearly explained.
· Similarly, the design for data presented in Figure 6 is not straightforward. If both groups were stressed, it is inappropriate to name one of them as “control”.
· A leap of logic is found for results in figures 6 and 7. There are both excitatory and inhibitory neurons in the lateral septum, but no evidence was provided to unambiguously show whether binge eating increased activity in both populations or one of them. Without a clear clue, it is not justifiable to simply activate the GAD2-expressing interneurons in this region. Moreover, did chronic stress per se increase activation of GAD2+ interneurons in the lateral septum?
· Chemogenetic activation is not physiological and may lead to artifacts. Considering that chronic stress per se increased anxiety, it would be more reasonable to inhibit the activity of GAD2+ neurons in the lateral septum, instead of activating them.
· After a careful check of the data, it seems that social defeat stress did not change anxiety level in mCherry-expressing mice. Why there is an absence of stress effects?
Minor issues:
· The reviewer is confused about the unit for exploration speed (check figure 1C for example). Did mice only move 1 millimeter per second on average (like a snail)? According to the reviewer’s experience, cm/s should be m/s.
· Figure 2, to make the naming of groups consistent, the CUMS groups should be named as CUMS+water, which was used for the following figures.
· For data shown in figure 3, one-way repeated measures ANOVA should be used instead.
· Figure 6A, what do the heat map represent for? Cell density or optical density? The full name for the abbreviations should be provided in the figure legend.
· For figure 7, what is the duration of the social defeat stress? The defeat stress procedure should be described in the methods.
· The duration of each behavioral task (e.g., OFT) should be clearly specified. Ideally, the illumination level in anxiety tests should also be stated.
· The post hoc multiple comparison method for ANOVA should be specified.
· The ANY-maze v3.0 is a very old version. Please make sure you are indeed using it.
Author Response
Reviewer 1
Major issues:
- Strictly speaking, control groups in many experiments are missing, e.g., the two unstressed groups in figures 2-7. The rationale for using such a simplified experimental design should be clearly explained.
R1: We agree with you that it would be better if we designed unstressed control groups in figures 2-7. Our preliminary data showed sucrose solution intake did not alter the baseline anxiety level in the open-field tests, and it may be due to the detection sensitivity of the paradigm we used. We further checked the effects of one-week CUMS administration, which did not display anxiety-like behavior. We then checked the changes of the anxiety level in these one-week CUMS-treated mice with sucrose solution intake. We found that the basal anxiety-level of these mice did not change. Therefore, we speculate that the sucrose solution exerts its relief effect only on the anxious individuals, but has no significant effect on the normal/basal anxiety level. In this study, we thus focus on the role of sucrose in regulation of stress-induced anxiety, and then we designed two groups paired with stress. Thank you again for your comments.
- Similarly, the design for data presented in Figure 6 is not straightforward. If both groups were stressed, it is inappropriate to name one of them as “control”.
R2: Thank you for pointing this out. We have revised the “control” to “CUMS” in Figure 5 and 6 accordingly.
- A leap of logic is found for results in figures 6 and 7. There are both excitatory and inhibitory neurons in the lateral septum, but no evidence was provided to unambiguously show whether binge eating increased activity in both populations or one of them. Without a clear clue, it is not justifiable to simply activate the GAD2-expressing interneurons in this region. Moreover, did chronic stress per se increase activation of GAD2+ interneurons in the lateral septum?
R3: We agree with your opinion very much. According to the Allen Brain and Sheehan’s work (Brain Res Brain Res Rev ,2004), almost all the neuron in LS (> 90%) were found to be GABAergic. And in humans, variations in glutamate decarboxylase 2 (GAD2) are associated with feeding behaviors, emotional eating, and weight gain (Choquette, A.C. et al, Physiol Behav, 2009). We therefore want to figure out the role of these LS GABAergic neurons in this binge phenotype by using GAD-2-cre mice. In our further study, experiments should be done to detect the functions of the excitatory neurons in vLS. Moreover, current evidence suggests that vLS was activated by social-defeat in female mice (Newman and Covington et al., Biol Psychiatry, 2019), implying the important roles of the GAD2+ interneurons in the lateral septum in the chronic stress group.
- Chemogenetic activation is not physiological and may lead to artifacts. Considering that chronic stress per se increased anxiety, it would be more reasonable to inhibit the activity of GAD2+ neurons in the lateral septum, instead of activating them.
R4: We agree with your comments very much. We found that acute chemogenetic activation vLSGAD2 induced compulsive eating test and anxiety-like behavior. As your suggestion, it makes more sense that chronic inhibit the activity of vLSGAD2. Interestingly, a recent study found that activation of LS Glp1r neurons is required for acute restraint stress-induced hypophagia, and they inhibiting LS Glp1r neurons using chemogenetic tools completely blocked stress-induced-hypophagia (Bales and Centanni et al., Mol Metab, 2022). Their findings implied the vLSGAD2 neurons may play the same role in binge behavior. In the future study, we will design a long-term chemogenetic inhibition of vLSGAD2 neurons to mimic anorexia or loss of appetite and observe weight change.
- After a careful check of the data, it seems that social defeat stress did not change anxiety level in mCherry-expressing mice. Why there is an absence of stress effects?
R5: Thank you for pointing out this. Studies (Li and Xiang et al., J Neuroinflammation, 2022) and our parallel study demonstrated that acute social defeat stress did induce anxiety-like behavior and serum corticosterone increasing. In this study, there was no apparent anxiety-like behavior after stress in OFT between these two batches. But from the EPM data, we found that open-arm entries were significantly less in the stress group compared with naïve mCherrys-control (Figure S1 A-B), which indicated that mice exhibited higher level of anxiety after social defeat stress. But after chemogenetic activation vLSGAD2, EPM test showed no difference between these two groups (Figure S1 C-D). We hypothesized that vLSGAD2 neurons activation did not alter the anxiety level, but instead induced compulsive eating behavior. We reorganized the supplementary figure 1 and please refer to new Figure S1.
Minor issues:
- The reviewer is confused about the unit for exploration speed (check figure 1C for example). Did mice only move 1 millimeter per second on average (like a snail)? According to the reviewer’s experience, cm/s should be m/s.
R6: As suggested, we corrected the unit to m/s.
- Figure 2, to make the naming of groups consistent, the CUMS groups should be named as CUMS+water, which was used for the following figures.
R7: Thanks for your excellent advice, we re-named the groups as your suggested.
- For data shown in figure 3, one-way repeated measures ANOVA should be used instead.
R8: Thanks for your excellent advice, we re-analyzed as your suggested, and fix the analysis result in manuscript.
- Figure 6A, what do the heat map represent for? Cell density or optical density? The full name for the abbreviations should be provided in the figure legend.
R9: Thanks for your comments. Heat map represents for the c-Fos positive cells in each brain area (Cell density). And the full name for the abbreviations was provided in the figure legend as your suggestion.
Manuscript line 210-223: “Abbreviations: AOM anterior olfactory nucleus medial part, IL infralimbic cortex, DTT dorsal tenia tecta, SHi septohippocampal nucleus, PrL prelimbic cortex, dLS lateral septal nucleus dorsal part,NAcC accumbens nucleus core, NAcSh accumbens nucleus shell, iLS lateral septal nucleus intermediate part, vLS lateral septal nucleus ventral part, BSTL bed nucleus of the stria terminalis lateral division, BSTMV bed nucleus of the stria terminalis medial division ventral part, PVA paraventricular thalamic nucleus anterior part, BLA basolateral amygdaloid nucleus anterior part, MD mediodorsal thalamic nucleus, CM central medial thalamic nucleus, Re reuniens thalamic nucleus, AHC anterior hypotha-lamic area central part, Pa paraventricular hypothalamic nucleus, PVT paraventricular thalamic nucleus, VMH ventromedial hypothalamic nucleus, SubI subincertal nucleus, DM dorsomedial hypothalamic nucleus, LHb lateral habenular nucleus, PH posterior hypothalamic area, PSTH parasubthalamic nucleus, SPF subparafascicular thalamic nu-cleus, VLGMC ventral lateral geniculate nucleus magnocellular part, VTA ventral teg-mental area.”
- For figure 7, what is the duration of the social defeat stress? The defeat stress procedure should be described in the methods.
R10: Thanks for your comments. We have revised the methods accordingly.
Manuscript line 375-377: “For stress challenge in GAD2-Cre mice, aggressive CD1 mice co-house with clear acrylic sheet to separate the C57BL/6 and CD1 mice for 24 h, and behavior test was operated 30min later.”
- The duration of each behavioral task (e.g., OFT) should be clearly specified. Ideally, the illumination level in anxiety tests should also be stated.
R11: The duration of OFT, EPM and CPP was 300s under lamp,the duration of binge and compulsive eating test was 60min. We have revised the methods accordingly.
- The post hoc multiple comparison method for ANOVA should be specified.
R12: Post hoc multiple comparison method for ANOVA was Tukey or Bonferroni. We have revised the methods accordingly.
Manuscript line 459-460: “Post hoc multiple comparison method for ANOVA was Tukey or Bonferroni.”
- The ANY-maze v3.0 is a very old version. Please make sure you are indeed using it.
R13: Sorry for that clerical error, we have updated our ANY-maze to v7.1.
New version of manuscript was in the attachment.

Reviewer 2 Report
In this article, the authors described in great detail the procedures and the rationale for investigating the prevention of stress-induced anxiety by sugar. Moreover, the authors show that sugar causes compulsive eating. This article is very interesting and many details were described. In my opinion, minor considerations need to be addressed before this article is published.
-The effect of sugar on stress should be explained more in the introduction in 2-3 sentences.
-The first sentence of the conclusion should be deleted. It seems it is a mistake.
-It is better for the authors to explain the heat map of the representative mice.
-What does the behind mean in “behind sugar” and “behind water” in CPP? It should be explained.
Author Response
Reviewer 2
- The effect of sugar on stress should be explained more in the introduction in 2-3 sentences.
R1: Thanks for your comments. We have revised the introduction accordingly.
Manuscript line 48-52: “Stress has clear effects on mood including increased anxiety, for many individuals, stress promotes the consumption of highly palatable and high-sugar foods in response to negative emotions such as anxiety [3]. Consuming high-sugar foods can attenuate the psycho-logical (anxiety and depressive mood) and physiological effects of stress”.
- The first sentence of the conclusion should be deleted. It seems it is a mistake.
R2: Thanks for your comments. We have revised the conclusion accordingly.
- It is better for the authors to explain the heat map of the representative mice.
R3: Thanks for your comments. We have revised the conclusion accordingly.
Manuscript line 121-127:” The heat map of the OFT representative mice was shown as Figure 2E, red color means mice spent more time in that place.” “The heat map of the EPM representative mice was shown as Figure 2I, red color means mice spent more time in that place.”
- What does the behind mean in “behind sugar” and “behind water” in CPP? It should be explained.
R4: We used a home cage-like box to measure the time spent of mice in four quadrants, a quadrant with sucrose, a quadrant with water. According to your comments, the other two empty quadrants could be re-named as “No Sucrose” and “No Water”.
Manuscript line 163-164: “blank quadrant no sucrose (NS), and blank quadrant no water (NW).”
New version of manuscript was in the attachment.
